# LevAttention: Time, Space and Streaming Efficient Algorithm for Heavy Attentions

**Ravindran Kannan**[*]
Simons Institute, UC Berkeley
kannan100@gmail.com

**Chiranjib Bhattacharya**
Indian Institute of Science
chiru@iisc.ac.in

**Praneeth Kacham**
Google Research
pkacham@google.com

**David P. Woodruff**[†]
Carnegie Mellon University
dwoodruf@cs.cmu.edu

## Abstract

A central problem related to transformers can be stated as follows: given two $n \times d$ matrices $Q$ and $K$, and a non-negative function $f$, define the matrix $A$ as follows: (1) apply the function $f$ to each entry of the $n \times n$ matrix $QK^T$, and then (2) normalize each of the row sums of $A$ to be equal to 1. The matrix $A$ can be computed in $O(n^2 d)$ time assuming $f$ can be applied to a number in constant time, but the quadratic dependence on $n$ is prohibitive in applications where it corresponds to long context lengths. For a large class of functions $f$, we show how to find all the "large attention scores", i.e., entries of $A$ which are at least a positive value $\varepsilon$, in time with linear dependence on $n$ (i.e., $n \cdot \text{poly}(d/\varepsilon)$) for a positive parameter $\varepsilon > 0$. Our class of functions include all functions $f$ of the form $f(x) = |x|^p$, as explored recently in transformer models. Using recently developed tools from randomized numerical linear algebra, we prove that for any $K$, there is a "universal set" $U \subset [n]$ of size independent of $n$, such that for any $Q$ and any row $i$, the large attention scores $A_{i,j}$ in row $i$ of $A$ all have $j \in U$. We also find $U$ in $n \cdot \text{poly}(d/\varepsilon)$ time. Notably, we (1) make no assumptions on the data, (2) our workspace does not grow with $n$, and (3) our algorithms can be computed in streaming and parallel settings. We call the attention mechanism that uses only the subset of keys in the universal set as LevAttention since our algorithm to identify the universal set $U$ is based on leverage scores. We empirically show the benefits of our scheme for vision transformers, showing how to train new models that use our universal set while training as well, showing that our model is able to consistently select "important keys" during training. We also provide theoretical motivation by formulating a planted model in which our efficient algorithms provably identify relevant keys for each query.

## 1 Introduction

A transformer architecture is one of the most popular architectures for building foundation models, with applications to natural language processing, computer vision, and many other modalities and their combinations. It is well-known that exact computation of their attention layers naïvely requires quadratic (in the context length) time, which poses a huge problem for scalability. A large body of work has tried to improve the efficiency of computing attention layers under a variety of assumptions, including imposing sparsity constraints (Parmar et al., 2018; Child et al., 2019; Beltagy et al., 2020; Kitaev et al., 2020; Tay et al., 2020), kernel methods (Bello et al., 2021; Choromanski et al., 2021; Peng et al., 2021; Zheng et al., 2022), low rank assumptions (Wang et al., 2020; Xiong et al., 2021; Ma et al., 2021), and assumptions of bounded entries or conditions on column or row norms (Alman & Song, 2023; Han et al., 2024).

---

[*]R. Kannan is the first author and the remaining authors are ordered alphabetically.

[†]Part of this work done while D. Woodruff was at Google Research and at the Simons Institute for the Theory of Computing. D. Woodruff also acknowledges Office of Naval Research award number N000142112647.

In each attention layer, one receives as input an $n \times d'$ input matrix $X$, which may be the embedding of the tokenization of the input, or an input from previous layers in the transformer. Here $n$ is the context length and $d'$ is the embedding dimension of each token, which is typically much smaller than $n$. From $X$ we multiply by three $d' \times d$ learned matrices $W^Q$, $W^K$, and $W^V$, and define the query matrix $Q = X \cdot W^Q$, the key matrix $K = X \cdot W^K$, and the value matrix $V = X \cdot W^V$. One then outputs the attention matrix, which is defined to be $D^{-1} \cdot f(QK^T) \cdot V$, where $D$ and $f(QK^T)$ are each $n \times n$ matrices defined as follows. For a non-negative function $f$, we apply $f$ entry-wise to the $n \times n$ matrix $QK^T$ to form $f(QK^T)$. Then we let $D$ be a diagonal matrix with $D_{i,i} = \sum_{j=1}^{n} f(\langle Q_i, K_j \rangle)$, where $Q_i$ is the $i$-th row of $Q$ and $K_j$ is the $j$-th row of $K$, respectively. Let $A = D^{-1} f(QK^T)$. Note that the entries of each row of $A$ are non-negative and sum to 1, and each row of $A \cdot V$ can be viewed as a non-negative combination of the rows of $V$, where the coefficients of the combination sum to 1.

Instantiating the above framework with $f(x) = e^{x/\sqrt{d}}$ corresponds to taking a softmax of each row of $QK^T/\sqrt{d}$ and then multiplying by $V$. In this case, by appropriately scaling the hard instance in Alman & Song (2023), one can show that computing attention with high precision requires $n^{2-o(1)}$ time under a standard complexity-theoretic assumption. In an attempt to bypass this hardness, recent work has considered replacing $f$ with other functions, such as $f(x) = x^p$ for a positive even integer $p$. Indeed, both the PolySketchFormer (Kacham et al., 2024) as well as the work of Sarlós et al. (2023) consider using TensorSketch to speed up the computation of an approximate attention matrix for such functions $f$. Motivated by these works, we define the $f$-sensitivities $\sigma_i^f$ of an $n \times d$ matrix $K$ for $i = 1, 2, \ldots, n$ as follows:

$$\sigma_i^f(K) = \sup_{y \neq 0} \frac{f(\langle K_i, y \rangle)}{\sum_{j=1}^{n} f(\langle K_j, y \rangle)}.$$

When $f(x) = x^2$, these are just the so-called *leverage scores* of the rows of matrix $K$, which are well-studied in randomized numerical linear algebra (see, e.g., Mahoney et al. (2011); Woodruff et al. (2014)). For general $p$, these are known as the $\ell_p$-sensitivities, which are also well-studied (see, e.g., Woodruff & Yasuda (2023); Padmanabhan et al. (2023)), and they can be bounded by the so-called $\ell_p$-Lewis weights of the matrix $K$ (Cohen & Peng, 2015) (for background, see, e.g., Section 3.3 of Clarkson et al. (2019)). Many interesting properties of such scores are known. One such property is the following. Let $\Psi^f = \sup_K \sum_{i=1}^{n} \sigma_i^f(K)$. When $f(x) = |x|^p$ for $1 \leq p \leq 2$, one has $\Psi^f \leq d$, and when $f(x) = |x|^p$ for $p \geq 2$, one has $\Psi^f \leq d^{p/2}$. These bounds do not depend on the context length $n$.

Critical to our work will be the observation in practice that the matrix $A$ is often well-approximated by retaining only its large entries, i.e., preserving all entries above a certain threshold $\varepsilon > 0$ and replacing the remaining entries with $0$, or perhaps fitting a low rank approximation to the remaining entries (Gupta et al., 2021; Wang et al., 2022). The entries of $A$ are called the attention scores or attention weights, and we will say a score is *large* if its value is at least $\varepsilon$.

## 1.1 OUR CONTRIBUTIONS

We outline our contributions below.

**Existence of a Universal Set.** We prove that for a large class of functions $f$, for any key matrix $K$, there is a small "universal set" $U \subset [n] = \{1, 2, \ldots, n\}$ of size independent of $n$, such that for any query matrix $Q$ and any $i \in \{1, 2, \ldots, n\}$, the large attention scores $A_{i,j}$ in row $i$ of $A$ all have $j \in U$. One of our results, which combines some ideas from Sections 2 and 3, is the following:

**Theorem 1.1.** *Let $f$ be a non-negative function and let $\Psi^f = \sup_K \sum_{i=1}^{n} \sigma_i^f(K)$. There is a subset $U \subset [n]$ of size $\Psi^f/\varepsilon$ so that for any query matrix $Q$ and $i \in \{1, 2, \ldots, n\}$, if $A_{i,j} \geq \varepsilon$, then $j \in U$.*

We note that for $f(x) = x^2$, the $f$-sensitivities are just the leverage scores of the matrix $K$, and it is known that $\Psi^f = d$ (see, e.g., Mahoney et al. (2011); Woodruff et al. (2014)). In this case, by Theorem 1.1, we have $|U| = d/\varepsilon$. More generally, if $f(x) = |x|^p$, we have $\Psi^f \leq d$ for $1 \leq p \leq 2$ and $\Psi^f \leq d^{p/2}$ for $p > 2$ (see, e.g., Section 3.3 of Clarkson et al. (2019)), and so $|U| \leq \max(d, d^{p/2})/\varepsilon$. Additional bounds are known for other functions $f$, e.g., from the work of

Musco et al. (2022). There, for example, if $f(x)$ is the Huber function $f(x) = x^2$ for $|x| \leq \tau$ and $f(x) = |x|$ for larger $|x|$, then $\Psi = O(d \log n)$, or if $f(x)$ is the Tukey function $f(x) = x^2$ for $|x| \leq \tau$, and $f(x) = \tau$ for larger $|x|$, then $\Psi = O(d \log n)$. Here $\tau > 0$ is any specified threshold.

We give an outline of the proof of Theorem 1.1 here. If $A_{i,j}$ is a large attention score, then $A_{i,j} = \frac{f(\langle Q_i, K_j \rangle)}{\sum_{\ell} f(\langle Q_i, K_\ell \rangle)} \geq \varepsilon$. It follows that $\sup_y \frac{f(\langle y, K_j \rangle)}{\sum_{\ell} f(\langle y, K_\ell \rangle)} > \varepsilon$ and so $\sigma_i^f(K) \geq \varepsilon$. If we define $U$ to be the set of $i$ for which $\sigma_i^f(K) \geq \varepsilon$, then $i \in U$ and we have $|U| \cdot \varepsilon \leq \Psi^f(K)$, and so $|U| \leq \Psi^f(K)/\varepsilon$.

We stress that our set $U$ does not depend on any particular query or query matrix $Q$, i.e., for any possible future query $q$, any large attention scores it participates in necessarily involve keys in $U$.

**Fast Algorithm for Finding $U$.** There are very efficient algorithms for computing the set $U$ for many interesting functions $f$. For example, when $f(x) = x^2$, these are the large leverage scores and a simple way of computing them is by computing a QR-decomposition of the matrix $K$ and finding all rows with squared norm at least $\varepsilon$. This takes $O(nd^2)$ time and has the advantage of computing the leverage scores exactly, resulting in the smallest possible set $U$. There are also sketching techniques for more quickly finding the large leverage scores (Drineas et al., 2012; Clarkson & Woodruff, 2013) in time $\mathrm{nnz}(K) + \mathrm{poly}(d/\varepsilon)$, up to logarithmic factors, where nnz denotes the number of non-zero entries of $K$. Similarly, for $f(x) = |x|^p$ one can use the $\mathrm{nnz}(K) + \mathrm{poly}(d/\varepsilon)$ time algorithm for finding $\ell_p$-Lewis weights in Cohen & Peng (2015), and the fact that a scaling of these weights bounds the $f$-sensitivities (see, e.g., Section 3.3 of Clarkson et al. (2019)).

**Time and Memory-Efficient Algorithm for Finding Large Attentions Given $U$.** We do not make any assumptions on the data and our theorem holds for *any query* matrix $Q$, and thus given any potentially new query $q$, one only needs to search in the set $U$ for the set of large attention scores involving $q$. Thus, instead of naïvely spending $O(nd)$ time to walk through each possible key to find the large attention scores, one only needs to spend $O(\Phi^f d/\varepsilon)$ time, assuming that $f$ can be evaluated in constant time. Notice that our time per query does not grow with the context length $n$. Further, as we only store the keys in the set $U$, our workspace also does not grow with the context length.

We also show how, for $f(x) = x^p$ for even integers $p$ it is possible to spend at most $n \cdot \mathrm{poly}(d)$ time preprocessing $K$ so that from $U$ and $K$, given a query $q$, one can output the *exact* value of all large attention scores involving $q$ in $\mathrm{poly}(d/\varepsilon)$ time. Note that this is not as trivial as simply computing $\langle q, K_i \rangle^p$ for each $K_i \in U$, since we also would like to compute the exact normalization factor in the attention matrix, and do not want to spend $n$ time to do so. For $p = 2$ this amounts to computing $\|Kq\|_2^2$, but by computing the SVD of $K$ in a preprocessing step, this quantity can be computed in only $O(d^2)$ time. For even integers $p > 2$, we can reduce to the case of $p = 2$ by using Khatri-Rao products on the rows of $K$, as well as Khatri-Rao products of the query with itself. One can further speed up these computations by approximating the heavy attention values using sketching.

**Extension to Streaming Environments.** It is also possible to compute the set $U$ in a streaming environment. We illustrate the idea for $f(x) = x^2$, though using the tensor trick discussed above, it also applies to $f(x) = x^p$ for any even integer $p$. The simplest algorithm is a two-pass algorithm over $K$, where we maintain the $d \times d$ matrix $K^T K$ by summing $n$ outer products as we read each of the $n$ keys. Then, in a second pass, the $i$-th leverage score is $K_i^T (K^T K)^{-1} K_i$, and we simply let $U$ be the set of keys for which the leverage score is at least $\varepsilon$. The total memory is only $O(d^2/\varepsilon)$.

More interestingly, one can compute $U$ in a single pass over $K$ by again maintaining $K^T K$ but also storing all keys whose *online leverage score* Cohen et al. (2016a) is at least $\varepsilon$. It is known that the online leverage scores are at least the leverage scores and sum to $O(d \log \kappa^{OL})$, where $\kappa^{OL}$ is the online condition number, which can be bounded by $n^{O(d)}$ assuming the entries of $K$ have $O(\log n)$ bits of precision. In this case, at the end of the stream, one simply evaluates $K_j^T (K^T K)^{-1} K_j$ for each key $K_j$ that was stored because it had a large online leverage score. The total memory is $\mathrm{poly}(d/\varepsilon)$ and the algorithm is a single pass algorithm.

**Extension to Distributed Environments.** It is possible to find $U$ in a distributed environment, where the keys are distributed across multiple servers. In this case if server $i$ holds an $n_i \times d$ matrix $K^i$, so that $K = [K^1; K^2; \ldots, K^s]$ if there are $s$ servers, then the $i$-th server can communicate $(K^i)^T K^i$ to the first server for each $i$, and the first server can add these up to compute $K^T K$ and send $K^T K$ to each server. Then the $i$-th server can send a set $U^i$ of keys $K_j$ that it holds for which

$K_j^T (K^T K)^{-1} K_j \geq \varepsilon$. Then $U = \cup_j U_j$. Although our discussion has focused on $f(x) = x^2$, similar ideas exist for $f(x) = x^p$ for any $p$, as well as the functions studied in Musco et al. (2022).

**Planted Model.** Finally, we formulate a planted model of keys and queries, where each query is a noisy linear combination of a small subset of keys. We show that in this model, we can find all keys relevant to a given query in sublinear time. We give deterministic conditions under which this is possible, and describe a natural stochastic setting where these conditions are realized.

**Experiments.** We perform experiments on pretrained ViT models to empirically understand the structure of the attention matrices that arise for typical inputs to a model. Our experiments suggest that for many attention heads, a small subset of *universal* keys along with a set of local tokens capture a large fraction of attention weight for a large number of tokens. This motivates the identification problem that we study in this paper.

We then evaluate the effectiveness of leverage score selection for the downstream task of image classification using the pretrained softmax model. In this experiment, at each attention head, we compute a subset of keys with the largest leverage scores and make the queries attend only to that subset of keys. We observe that while the accuracy of the model drops compared to the full attention model, the model still maintains non-trivial accuracy and that leverage score selection obtains better accuracy compared to e.g., row norm based selection, showing that leverage score based selection does compute a subset of important keys.

We also trained multiple ViT models from scratch using the leverage score based attention mechanism and observe that the model quality improves significantly compared to doing inference on the softmax pretrained models using the leverage score mechanism. Across all the models, we observe that the model quality achieves >90% accuracy of the full softmax attention while selecting the top 32 keys (out of 197 keys for L/16 and S/16 models and out of 785 keys for the L/8 model) using the leverage score mechanism at each attention head. On the negative side, we observe that models trained from scratch using squared row norm selection or even "random key selection", where we select a set of random key positions for each attention head and each query attends to only keys in those positions throughout the training/inference process, attains similar accuracies. However, unlike leverage score sampling, these latter methods do not have the "universality" property as discussed in the introduction for a relaxed definition of the attention mechanism. We leave it as an open question how to make effective use of leverage score information to train better models.

**Notation:** For a matrix $B$, we use $\mathrm{nnz}(B)$ to denote the number of non-zero entries of $B$. We let $\omega$ be the exponent of matrix multiplication, so that multiplying two $d \times d$ matrices takes $d^\omega$ time.

## 2 CONNECTION TO LEVERAGE SCORES

We define the Generalized Attention Problem (GAP): let $n$, and $D \geq d$ be positive integers. Consider two known maps: $\psi, \phi \colon \mathbb{R}^d \to \mathbb{R}^D$. We assume that $\phi$ and $\psi$ are computable in time $O(D)$. We are given matrices $Q$ and $K$, where $Q$ is $n \times d$ and $K$ is $n \times d$. GAP is the problem of computing the $n \times n$ matrix $A$ defined by $A_{ij} = \frac{\langle \psi(Q_i), \phi(K_j) \rangle^2}{\sum_{\ell=1}^n \langle \psi(Q_i), \phi(K_\ell) \rangle^2}$, where $K_j$ denotes the $j$-th row of $K$.

**Remark 2.1.** *One special case is the SoftMax operation. In this case the value $D$ is infinite, but Softmax can be approximated with finite $D$, see, e.g., Choromanski et al. (2021). Note that $\phi = \psi$ in approximations to Softmax. Another set of interesting special cases occurs when $\phi$ and $\psi$ are polynomial maps, as in Sarlós et al. (2023); Kacham et al. (2024).*

GAP requires $\Omega(n^2)$ time to write down the entries of $A$. We show how to improve this by finding only the large entries of $A$, i.e., those that are at least $\varepsilon$.

**Theorem 2.1.** *There is a deterministic algorithm $\mathcal{A}$ that finds a subset $U$ of rows of $K$ (i.e., $U \subseteq [n]$) satisfying: $|U| \leq \frac{D}{\varepsilon}$ and for all $Q_{n \times d}$ and for all $i \in \{1, 2, \dots, n\}$: $\{j \colon A_{ij} \geq \varepsilon\} \subseteq U$.*

We show that it suffices to prove this for the special case $D = d$ and $\phi, \psi$ are the identity maps.

**Lemma 2.2.** *Without loss of generality, we can assume in Theorem 1 that $D = d$ and $\phi, \psi$ are both the identity map (i.e., $\phi(x) = \psi(x) = x$).*

*Proof.* The proof follows from a kernel-type construction. From $K$, define an $n \times D$ matrix $\widehat{K}$ given by $\widehat{K}_j = \phi(K_j)$. For any $n \times d$ matrix $Q$, define an $n \times D$ matrix $\widetilde{Q}$ by $\widetilde{Q}_j = \psi(Q_j)$. Further, define an $n \times n$ matrix $\widehat{A}$ by: $\widehat{A}_{ij} = \frac{\langle \widetilde{Q}_i, \widehat{K}_j \rangle^2}{\sum_{\ell=1}^n \langle \widetilde{Q}_i, \widehat{K}_\ell \rangle^2}$. It is easy to see that $\widehat{A} = A$. Assuming that Theorem 2.1 holds when $D = d$ and $\phi, \psi$ are both the identity map, we can find a $\widehat{U}$ with $\{j \colon \widehat{A}_{ij} \geq \varepsilon\} \subseteq \widehat{U}$. Since $\widehat{A} = A$, we get that $\{j \colon A_{ij} \geq \varepsilon\} \subseteq \widehat{U}$, proving the lemma. $\qquad\square$

Restricting to the case when $D = d$ and $\phi(x) = \psi(x) = x$, we formally define the set $LA$ for large attentions scores: $LA(K, \varepsilon, v) = \{j \colon \langle K_j, v \rangle^2 \geq \varepsilon \sum_{\ell=1}^n \langle K_\ell, v \rangle^2\}$ for $v \in \mathbb{R}^d$. Let $U(K, \varepsilon) = \bigcup_{v \in \mathbb{R}^d \setminus \{0\}} LA(K, \varepsilon, v)$. The following is a simple characterization of $U(K, \varepsilon)$.

**Theorem 2.3.** *Let $U(K, \varepsilon)$ be the set of rows of $K$ with leverage score $\geq \varepsilon$, then $|U(K, \varepsilon)| \leq d/\varepsilon$.*

*Proof.* Observe that $j \in U(k, \varepsilon)$ if and only if there is a non-zero vector $v$ for which $\frac{\langle K_j, v \rangle^2}{\sum_{\ell=1}^m \langle K_\ell, v \rangle^2} \geq \varepsilon$, which means that the the $j$-th $f$-sensitivity $\sigma_j^f(K) \geq \varepsilon$. Conversely, if $\sigma_j^f(K) \geq \varepsilon$, then there exists a non-zero vector $v$ for which $\frac{\langle K_j, v \rangle^2}{\sum_{\ell=1}^m \langle K_\ell, v \rangle^2} \geq \varepsilon$, and so $j \in U(K, \varepsilon)$. It is a well-known fact that for $f(x) = x^2$ that $\sigma_j^f(K)$ is precisely the $j$-th leverage score of the matrix $K$, see, e.g., the solution to problem 2.2 in Woodruff (2021). Note also that $j \in U(K, \varepsilon) \implies \tau(K, j) \geq \varepsilon \implies LS(K, j) \geq \varepsilon$. As the sum of leverage scores is equal to $d$ (see, e.g., Foster (1953)), we have $|U(K, \varepsilon)| \leq \frac{d}{\varepsilon}$. $\qquad\square$

Thus, to compute $LA$, we simply need to compute the rows $j$ of $K$ with leverage score at least $\varepsilon$. Theorem 2.1 now follows from Lemma 2.2 and Theorem 2.3.

**Streaming Algorithms.** The $j$-th leverage score of $K$ is equal to $K_j^T (K^T K)^{-1} K_j$ (see, e.g., Mahoney et al. (2011); Woodruff et al. (2014)), and so a simple 2-pass streaming algorithm using $O(d^2)$ memory would be to maintain $K^T K$ as a sum of $n$ outer products in a first pass over the keys of $K$, and then to compute $K_j^T (K^T K)^{-1} K_j$ exactly in the second pass over keys $K_j$, and store those $K_j$ for which this quantity is at least $\varepsilon$. This simple 2-pass streaming algorithm uses $O(d^2)$ words of memory and can be implemented in $O(nd^2 + d^\omega)$ time.

One can obtain a 1-pass algorithm to compute the rows $j$ of $K$ with leverage score at least $\varepsilon$ with only slightly more memory. The idea is to store the set $S$ of rows $K_j$ with *online leverage score* Cohen et al. (2016a) at least $\varepsilon$ in the first pass, and to also store $K^T K$. The $j$-th online leverage score is equal to $K_j^T ((K^{j-1})^T (K^{j-1}))^{-1} K_j$, where $K^{j-1}$ denotes the prefix of the first $j - 1$ rows of $K$. The $j$-th online leverage score is at least the $j$-th leverage score since sensitivities cannot increase as more rows are added to $K$, but more interestingly, Theorem 2.2 of Cohen et al. (2016a) shows that the sum of online leverage scores is bounded by $O(d \log \kappa^{OL})$, where $\kappa^{OL}$ is the online condition number, see Lemma 3.3 of Woodruff & Yasuda (2022). It is well-known for matrices with integer entries bounded by $\mathrm{poly}(n)$, which is an assumption often used to obtain meaningful memory bounds in a data stream, that the online condition number is at most $n^{O(d)}$ (see, e.g., Lemma 4.1 of Clarkson & Woodruff (2009) which lower bounds the minimum non-zero singular value of an $n \times d$ such matrix by $n^{-O(d)}$), and thus the number of rows stored will be at most $O(d^2 \log n)$, and so the memory is bounded by $O(d^3 \log n)$ words. One can also maintain $K^T K$ in the stream as before. At the end of the stream, for each row $K_j$ stored which had online leverage score at least $\varepsilon$, one can compute its exact leverage score $K_j^T (K^T K)^{-1} K_j$ at the end of the stream in order to exactly find LA. Overall this is a 1-pass algorithm with memory $\mathrm{poly}(d \log n)$ words.

**Finding all Heavy Attentions for a Query.** Given a query $q$, for $f(x) = x^2$ we can compute all large attention score values it participates in *exactly* because we can first preprocess $K$ in $O(nd^2)$ time to compute its singular value decomposition (SVD) $K = U' \Sigma V^T$. Then, given a query $q$, we can compute $\langle q, K_i \rangle^2$ for each $K_i \in U$ in $|U| \cdot d = O(d^2/\varepsilon)$ time, and we can also compute the normalization $\|Kq\|_2^2 = \|\Sigma V^T q\|_2^2$ in $O(d^2)$ time since $\Sigma V^T$ is a $d \times d$ matrix. Thus, after an initial preprocessing of $n \cdot \mathrm{poly}(d/\varepsilon)$ time, we can compute all large attention score values involving a query exactly and in only $\mathrm{poly}(d/\varepsilon)$ time. Note that if one would like to instead approximate the large attention values up to a $1 + \varepsilon$ factor, instead of computing the SVD of $K$, one can use a random sketching matrix $S$ so that $\|SKq\|_2^2 = (1 \pm \varepsilon) \|Kq\|_2^2$. If one uses the Subsampled Randomized

Hadamard Transform for example, then this improves the $O(nd^2)$ time required to compute the SVD of $K$ to only $nd \cdot \mathrm{poly}(\log n/\varepsilon)$ time and incurring $1/\mathrm{poly}(n)$ failure probability, e.g., using the analysis of Cohen et al. (2016b). It is also possible to use CountSketch in $O(nd + \mathrm{poly}(d))$ time with a $1/\mathrm{poly}(d)$ failure probability Clarkson & Woodruff (2013).

This section was for $f(x) = x^2$, while the next section handles $f(x) = |x|^p$ for any $p \geq 1$.

## 3 ATTENTION SCORES USING $f(x) = |x|^p$

We next consider $f$-sensitivities for $f(x) = |x|^p$.

**Theorem 3.1.** *Let $K$ be an $n \times d$ matrix. Let $p \in [1, \infty)$ and let $f(x) = |x|^p$. There exists a set $S$ of rows of $K$ containing all $f$-sensitivities at least $\varepsilon$ with the following properties:*

  *1. For $1 \leq p \leq 2$, $|S| = O(d/\varepsilon)$.*

  *2. For $p > 2$, $|S| = O(d^{p/2}/\varepsilon)$.*

*Moreover, $S$ can be computed in $(nnz(K) + poly(d/\varepsilon))poly(\log n)$ time.*

*Proof.* Fix $p \geq 1$ and let $f(x) = |x|^p$. Let $\tau_i$ denote the $\ell_p$-Lewis weights of $K$ (Cohen & Peng, 2015). These can all be computed up to a constant factor in total time $(nnz(K) + d^\omega)\mathrm{poly}(\log n)$ time (Cohen & Peng, 2015).

For $1 \leq p \leq 2$, it is known that $\tau_i$ upper bounds the $i$-th $f$-sensitivity $\sigma_i^f$. For $p > 2$, it is known that $d^{p/2-1}\tau_i$ upper bounds the $i$-th $f$-sensitivity. The $\ell_p$-Lewis weights sum to at most $d$. For an exposition of these statements, see e.g., Section 3.3 of Clarkson et al. (2019) and references therein.

Therefore, for $1 \leq p \leq 2$, the set $S$ of all rows $i$ with $\tau_i \geq \varepsilon$ satisfies the desired properties. For $p > 2$, the set $S$ of all rows $i$ with $d^{p/2-1}\tau_i \geq \varepsilon$ satisfies the desired properties. By approximating the $\tau_i$ up to a factor of 2 and including all $i$ whose approximate $\tau_i$ is at least $\varepsilon/2$ for $1 \leq p \leq 2$, while including all $i$ whose approximate $\tau_i$ is at least $\varepsilon/(2d^{p/2-1})$ for $p \geq 2$, we will include all rows $i$ whose $f$-sensitivity is at least $\varepsilon$. Further, if the approximate $\tau_i$ is at least $\varepsilon/2$, then the actual $\tau_i$ is at least $\varepsilon/4$, and so we have $|S| = O(d/\varepsilon)$ for $1 \leq p \leq 2$ while $|S| = O(d^{p/2}/\varepsilon)$ for $p > 2$.  □

We next show how to efficiently approximate the normalization term for all $1 \leq p < \infty$, in order to approximate all heavy attentions for a given query $q$ by using the set $U$. We will later show how to compute the normalization term exactly for $p$ an even integer.

**Theorem 3.2.** *Let $Q$ be an $n \times d$ matrix and $K$ be an $n \times d$ matrix. For $1 \leq p < \infty$ and any $j \in \{1, \ldots, n\}$, the normalization term $\sum_{i=1}^n |\langle Q_j, K_i \rangle|^p$ can be estimated efficiently. Specifically, there exists a sampling and rescaling matrix $S$ with the following properties:*

  *1. For $1 \leq p \leq 2$, $S$ has $d \cdot polylog(d/\varepsilon)/\varepsilon^2$ columns,*

  *2. For $p > 2$, $S$ has $d^{p/2}polylog/\varepsilon^2$ columns,*

  *3. With failure probability $\frac{1}{poly(d)}$, simultaneously for all $x \in \mathbb{R}^d$, $\|xK^TS\|_p^p = (1 \pm \varepsilon)\|xK^T\|_p^p$.*

*$S$ can be computed in $(nnz(K) + poly(d/\varepsilon))poly(\log n)$ time, and after this one-time computation, $\|xK^TS\|_p^p$ can be computed in $poly(d/\varepsilon)$ time.*

*Proof.* The normalization term can be written as $\|Q_jK^T\|_p^p$. We can obtain the matrix $S$ by Lewis weight sampling on $K^T$. This is a well-known technique for constructing subspace embeddings that preserve $\ell_p$ norms simultaneously for all $x$ Cohen & Peng (2015). Specifically, by Woodruff & Yasuda (2022), there exists an algorithm that computes a sampling and rescaling matrix $S$ with the stated number of columns such that for all $x \in \mathbb{R}^d$, $\|xK^TS\|_p^p = (1 \pm \varepsilon)\|xK^T\|_p^p$. The time to compute $S$ is $O(nnz(K) + poly(d/\varepsilon))poly(\log n)$. Once $S$ is computed, we can efficiently estimate

the normalization term for any $j$ as $\|Q_j K^T S\|_p^p$. Since $S$ has poly$(d/\varepsilon)$ columns, computing this term takes poly$(d/\varepsilon)$ time. □

**Finding all Heavy Attentions Exactly for a Query $q$.** Although the above procedure allows for quickly estimating the normalization term up to a $(1 + \varepsilon)$-factor, one may be interested in the exact value of the attention score if the model is sensitive to slight perturbations. One can also do this with a slightly larger time complexity for even integers $p > 2$, as the idea for $p = 2$ for computing heavy attention scores exactly extends to $f(x) = x^p$ for even integers $p$ using a tensor trick. Indeed, we can form the $n \times d^{p/2}$ matrix $K'$ by taking the Khatri-Rao product of each row of $K$ with itself $p/2$ times. Then we compute the SVD $K' = U'\Sigma V^T$. Given a query $q$, we can again compute $\langle q, K_i \rangle^p$ for each $K_i \in U$ in $|U| \cdot d = O(d^{p/2+1}/\varepsilon)$ time, and now we can also compute the normalization $\sum_j \langle K_j, q \rangle^p = \|K'q'\|_2^2 = \|\Sigma V^T q'\|_2^2$, where $q'$ is the Khatri-Rao product of $q$ with itself $p/2$ times. Thus, for constant even integers $p$, after an initial preprocessing of $n \cdot \text{poly}(d/\varepsilon)$ time, we can compute all large attentions for any particular $q$ exactly in only poly$(d/\varepsilon)$ time.

**Remark 3.1.** *The choice of $p$ in the definition of $f$-sensitivity can significantly impact the identification of large attention scores. Consider the following examples:*

**Case $p > 2$**: *Suppose one entry in a row of the attention matrix is $n^{1/p}$, while all other entries are $\Theta(1)$. This entry is a large attention with constant $\varepsilon$ for $f(x) = |x|^p$, but it is not a large attention score for $f(x) = x^2$ unless $\varepsilon$ is inverse polynomial in $n$.*

**Case $p < 2$**: *Suppose one entry in a row is $\varepsilon$, another entry is $1$, and the remaining entries are close to $0$. The entry with value $\varepsilon$ is a large attention score for $f(x) = x^2$ with value $\varepsilon^2$, but it is a large attention score for $f(x) = |x|$ with value $\varepsilon$. Thus, using $\ell_1$-sensitivities allows for faster identification and allows for storing a smaller set of large sensitivity rows of $K$.*

**Remark 3.2.** *In Musco et al. (2022), a method for efficiently computing $f$-sensitivities with respect to a generic function $f$ is presented. This function $f$ is assumed to satisfy basic properties such as subadditivity and symmetry. Examples include the Huber and Tukey loss functions. Musco et al. (2022) show that the sum of these sensitivities is $O(d \log n)$, implying the existence of a subspace embedding $S$ with poly$(d)$ columns to facilitate fast computation of the normalization factor. Furthermore, the universal set $U$ of rows of $K$ now has size $O(d \log n/\epsilon)$ and both $U$ and $S$ can be found in $O(nnz(K) + \text{poly}(d))\text{poly}(\log n)$ time. These generalized loss functions may offer more flexibility and different efficiency versus accuracy tradeoffs.*

## 4 A PLANTED MODEL

We formulate a planted model of keys and queries, where, each query is a noisy liner combination of a small set of keys; we call this set of keys, the "relevant keys for the query". We show that our algorithm, based on finding large attention scores, given $K$ and a query satisfying model assumptions, finds the relevant set of keys. For a plausible set of model parameters, the running time of our algorithm per query, amortized over many queries, is sublinear in $n$. Throughout, $K$ will be an $n \times d$ matrix. $K_j$ denotes the $1 \times d$ vector (the $j$-th row of $K$). Each row is a key. $q$ is a $1 \times d$ vector, and will denote a generic query. We assume there is a subset $S$ of keys such that for $i \in S$, the correlation of $K_i$ to other keys is upper bounded. We will assume later that query vectors are convex combinations of keys in $S$. We let $\delta_1, \delta_2$ be parameters satisfying $0 < \delta_2 \leq \delta_1 \leq 1/4$. We assume there is a subset $S$ of keys satisfying:

$$\forall j, \ell \in S, j \neq \ell, |K_j K_\ell^T| \leq \delta_1 \cdot \min(\|K_j\|_2^2, \|K_\ell\|_2^2) \tag{1}$$

$$\forall j \in S, \ell \notin S, |K_\ell K_j^T| \leq \delta_2 \cdot \min(\|K_j\|_2^2, \|K_\ell\|_2^2) \tag{2}$$

**Remark 4.1.** *$S$ may be thought of as a "stand out" subset of keys, since the correlation of $j \in S$ to other keys is upper bounded.*

We first argue that elements of $S$ have high self-attention scores.

**Lemma 4.1.** *Let $A = KK^T$. For all $i \in S$,* $\quad (A_{ii})^2 \geq \left(\sum_{j=1}^n (A_{ij})^2\right)/(1 + \delta_1^2|S| + \delta_2^2 n)$.

*Proof.*

$$(A_{ii})^2 = (K_i K_i^T)^2 = \|K_i\|_2^4.$$

$$\sum_j (A_{ij})^2 = |K_i K_i^T|^2 + \sum_{\ell \in S \setminus i} (K_\ell K_i^T)^2 + \sum_{\ell \notin S} (K_\ell K_i^T)^2 \leq \|K_i\|_2^4 \left(1 + \delta_1^2(|S| - 1) + \delta_2^2 n\right),$$

using (1) and (2). Now, both of them together imply the lemma. $\qquad\square$

Note that $\|K_i\|_2$ canceled out because of the normalization. This demonstrates the effectiveness of row normalization, which is a non-linear operation.

## 4.1 ASSUMPTION ON QUERIES

There is an unknown subset $S(q)$ [1] of $S$ and unknown weights $w_j(q)$ for $j \in S(q)$ satisfying

$$\sum_{j \in S(q)} w_j(q) \leq 1 \text{ and } w_j(q) \geq 4\delta_1 \text{ for all } j \in S(q) \tag{3}$$

and an unknown $d$-dimensional vector $z(q)$ [2] with

$$|z(q)K_\ell^T| \leq \frac{\delta_1}{4} \|K_\ell\|_2^2 \text{ for all } \ell \in [n], \text{ with } q = \sum_{j \in S(q)} w_j K_j + z(q). \tag{4}$$

We will just abbreviate $w_j(q)$ by $w_j$.

## 4.2 PROPERTY OF RELEVANT KEYS

**Theorem 4.2.** *We have $\forall j \in S(q) : qK_j^T \geq \frac{11}{4}\delta_1 \|K_j\|_2^2$ and $\forall j \notin S(q), qK_j^T \leq \frac{5}{4}\delta_1 \|K_j\|_2^2$.*

*Proof.* For $j \in S(q)$: $qK_j^T = w_j\|K_j\|_2^2 + \sum_{\ell \in S(q) \setminus j} w_\ell(K_\ell K_j^T) + z(q)K_j^T \geq \|K_j\|_2^2(4\delta_1 - \delta_1 - \delta_1/4) = (11/4)\delta_1\|K_j\|^2$, using (3), (4) and (1). For $j \in S \setminus S(q)$: $|qK_j^T| = \left|\sum_{\ell \in S(q)} w_\ell(K_\ell K_j^T) + z(q)K_j^T\right| \leq \|K_j\|_2^2(\delta_1 + (\delta_1/4)) = 5\delta_1\|K_j\|_2^2/4$, using (3), (4) and (1). For $j \notin S$: $|qK_j^T| = \left|\sum_{\ell \in S(q)} w_\ell(K_\ell K_j^T) + z(q)K_j^T\right| \leq \|K_j\|_2^2(\delta_2 + (\delta_1/4)) \leq (5/4)\delta_1\|K_j\|_2^2$ since $\delta_2 \leq \delta_1$. $\qquad\square$

## 4.3 ALGORITHM TO LEARN RELEVANT KEYS

Let $\frac{1}{1+\delta_1^2|S|+\delta_2^2 n} = \rho$. It is easy to see that $\frac{A_{ii}^2}{\sum_{j=1}^n A_{ij}^2}$ is at most the $i$-th leverage score of $K_i$ since the $i$-th leverage score is $\sup_{y \neq 0} \frac{\langle y, K_i \rangle^2}{\sum_{j=1}^n \langle y, K_j \rangle^2}$, which is at least as large as the value in this expression with the particular value $y = K_i$. Let $U = \{i : LS(K_i) \geq \rho\}$. Since the sum of all leverage scores is at most $d$, we have $|U| \leq d(1 + \delta_1^2|S| + \delta_2^2 n)$. For a suitable setting of parameters, we have $d(1 + \delta_1^2|S| + \delta_2^2 n) \in o(n)$ and $|U| \in o(n)$. Let $U' = \{i : \frac{(A_{ii})^2}{\left(\sum_{j=1}^n (A_{ij})^2\right)} \geq \rho\}$.

We can find $U'$ using our algorithm for finding heavy leverage scores. We then compute $\|K_i\|_2$ for all $i$. This takes $O(nd)$ time. To estimate the row lengths of $A = KK^T$, we use a Johnson-Lindenstrauss sketch to find the row lengths of $KK^T B$ for a random $n \times O(\ln n)$ Gaussian matrix $B$ (see, e.g., Woodruff et al. (2014) for background on Johnson Lindenstrauss sketches). When a query $q$ arrives, we check for each $i \in U'$ if $qK_i^T \geq (11/4)\|K_i\|_2^2$. This suffices since $S \subseteq U'$.

## 4.4 A STOCHASTIC EXAMPLE

We now present an example $K$ in which the assumptions (1) and (2) hold. The example has $K$ as a random matrix described below. There is a latent subspace $V$ of dimension $k \leq d/4$. For the conceptual description, we assume the first $k$ of the $d$ coordinates of a vector are in $V$ and the remaining $d - k$ coordinates are in $V^\perp$. Note that this is only for ease of description; we do not

---

[1] $S(q)$ is the set of keys relevant to query $q$.
[2] $z(q)$ can be thought of as noise.

actually know $V$. Intuitively, for $i \in S$, $K_i$ is mostly in $V$ with a small "noise" component in $V^\perp$, whereas, for $i \notin S$, $K_i$ is mostly in $V^\perp$ with a small component in $V$. The matrix drawn below describes the distributions from which each part is generated. The coordinates are i.i.d. inside each of the four parts with variance differing between parts. We assume $\delta_1 \geq (4/k)$ and $\delta_2 \geq (4\varepsilon_1/k)$. No condition on $\epsilon_0$ is required. By standard concentration bounds, (1) and (2) hold with high probability, so that Lemma (4.1) and Theorem (4.2) hold and our algorithm applies.

$$K_{n \times d} = \begin{bmatrix} \dfrac{K^1}{\mathcal{N}(0, 1/k)} & \dfrac{K^2}{\mathcal{N}(0, \varepsilon_1/(d-k))} \\ \dfrac{K^3}{\mathcal{N}(0, \varepsilon_1/k)} & \dfrac{K^4}{\mathcal{N}(0, 1/(d-k))} \end{bmatrix}$$

Here $K^1$ is $\varepsilon_0 n \times k$, and $K^2$ is $\varepsilon_0 n \times (d-k)$, and $K^3$ is $(1-\varepsilon_0)n \times k$, and $K^4$ is $(1-\varepsilon_0)n \times (d-k)$.

## 5 EXPERIMENTS

### 5.1 STRUCTURAL PROPERTIES OF TYPICAL ATTENTION MATRICES

We consider a pretrained ViT model Dosovitskiy et al. (2021) for image classification and consider the properties of the attention matrices across all of the attention heads in the model. The model we consider has six layers and each of the layers has six attention heads. The inputs to the model are tokenized into 197 tokens – 196 tokens representing the input image and an additional token whose final representation is used for classifying the image (see Figure 1). Therefore each of the attention matrices that we consider is of size $197 \times 197$.

For $i, j$, given the set of queries $\{Q_1, \ldots, Q_n\} \in \mathbb{R}^d$ and keys $\{K_1, \ldots, K_n\} \in \mathbb{R}^d$, then for $i \in [n]$ and $j \in [n]$, $A_{i,j} = \frac{\exp(\langle Q_i, K_j \rangle / \sqrt{d})}{\sum_j \exp(\langle Q_i, K_j \rangle / \sqrt{d})}$. For each $i$, we define $\ell_i(1), \ldots, \ell_i(n)$ such that $A_{i,\ell_i(1)} \geq \cdots \geq A_{i,\ell_i(n)}$.

**Top Heaviness.** We find that at many of the attention heads, the attention weight of a token is distributed in a "top heavy" manner, meaning that for many tokens $i$, the sum of the top-32 attention weights, defined as $\sum_{j=1}^{32} A_{i,\ell_i(j)}$ is quite large. We show the distribution of top-32 attention weights for an image input in Figure 2 in the Appendix. For a given attention head, we observe that the distribution of top-32 attention weights remains similar across multiple inputs.

**Locality.** We measure the amount of attention mass that is solely captured by the "neighboring" tokens (see Figure 1). We observe that only at a small fraction of heads, the attention mass of tokens is fully captured by the local tokens and that in general, a significant portion of the attention mass is distributed among non-local tokens. In Figure 3 in the Appendix, we show the histograms of the attention weights captured by local tokens at each of the attention heads in all the layers of the model for a typical input.

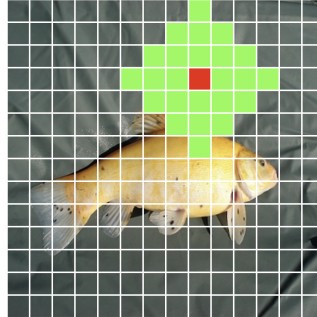

Figure 1: Example of an image partitioned into 196 patches using a 14 x 14 grid. The green patches represent the neighbors of the red patch within a distance of 3.

**Important Keys.** For $j \in [n]$, we define $W_j$ to be the non-local attention weight captured by $j$ as $\sum_{i:(i,j) \text{ are not neighbors}} A_{i,j}$. We define important keys to be the set of 32 keys with the largest $W_j$ values. We find that in the initial layers of the models, at many attention heads, most of the attention weight is captured by the local tokens and a small set of "important keys". In Figure 4 in the Appendix, we show the distribution of the attention weight captured by the set of "important keys" along with the local tokens.

### 5.2 INFERENCE FROM SOFTMAX ViT MODELS VIA LEVERAGE SCORE SELECTION

We consider three ViT models from the work of Dosovitskiy et al. (2021): (i) S/16, a small 22M parameter model that splits an image into 196 patches each of size $16 \times 16$ pixels, (ii) L/16, a large

| Model | Accuracy on validation set |
|---|---|
| S/16 (softmax) | 76.47% |
| S/16 (LevAttention, top-32, pretrained w/ softmax) | 13.3% |
| S/16 ($\ell_2$ norm selection, top-32, pretrained w/ softmax) | 3.3% |
| S/16 (LevAttention, top-32) | 68.30% |
| S/16 (LevAttention, top 64) | 72.48% |
| L/16 (softmax) | 78.83% |
| L/16 (LevAttention, top-32, pretrained w/ softmax) | 48.58% |
| L/16 ($\ell_2$ norm selection, top-32, pretrained w/ softmax) | 8.9% |
| L/16 (LevAttention, top-32) | 75.12% |
| L/16 (LevAttention, top-64) | 77.27% |
| L/16 (LevAttention, top-128) | 77.17% |
| L/8 (softmax) | 79.47% |
| L/8 (LevAttention, top-32, pretrained w/ softmax) | 0.8% |
| L/8 (LevAttention, top-32) | 71.96% |
| L/8 (LevAttention, top-64) | 74.64% |
| L/8 (LevAttention, top-128) | 76.69% |

Table 1: Accuracies of models with various attention mechanisms.

305M parameter model that splits an image into 196 patches each of size $16 \times 16$ pixels, and (iii) L/8, a large 305M parameter model that splits an image into 784 patches each of size $8 \times 8$ pixels. Note that all the models have one token appended whose representation after processing through the transformer is used for classifying images. We train the models on the Imagenet-1k (Russakovsky et al., 2015) dataset using the same hyperparameters as in the original work (Dosovitskiy et al., 2021). We find that the S/16, L/16 and L/8 models achieve accuracies of 76.47%, 78.83% and 79.47% respectively on the validation split of the Imagenet-1k dataset.

We then estimate the performance of leverage score selection on these datasets. At each attention head, we compute the keys with the 32 largest $\ell_2$ leverage scores and then make each query attend only to this set of keys in the attention mechanism. While we find that the accuracies drop significantly compared to the full softmax attention, our results show that the models still achieve non-trivial accuracies. In particular, the L/16 model has an accuracy of 48.58% using this mechanism. If we instead select top-32 keys based on the squared row norms, we observe that the accuracy of the L/16 model drops to 8.9%. This supports the idea that leverage scores are much better predictors of the *importance* of the keys compared to row norms.

### 5.3 TRAINING VIT MODELS

In the previous experiments, we used models trained using softmax attention and used LevAttention only at inference time. To see if the performance of the models improve if they are *aware* of LevAttention, we use the leverage score selection based attention mechanism to train the models as well. Using the same training setup as the softmax attention models, i.e., the same learning rate schedule, batch sizes, and optimizer, we see significant improvements in the validation accuracies. For the L/8 and L/16, we train for the initial 15% of the steps with full attention to obtain the warm start parameters and then train the remaining 85% of the steps using the leverage score selection based attention. We report the results in Table 1. To test if training the models with the awareness of "leverage score selection" truly is useful, we train models using (i) "row norm selection" where at each attention head, we pick 32 keys with the largest $\ell_2$ norms, and (ii) "random selection", where at each attention head, we pick 32 random key positions and only make the queries attend to those positions throughout the training process. We observe that these models achieve similar performance to the models trained via "leverage score selection". This shows that the "selection aware" training procedure is currently unable to translate the usefulness of leverage scores, as identified in the previous section, into obtaining better models than those achieved by "row norm selection" and "random selection" attention mechanisms. We note that only leverage scores have some provable guarantees, while the other methods do not. We leave the important question of obtaining better models trained using LevAttention as a future research direction.

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

# 6 APPENDIX: EXPERIMENTAL RESULTS

In this section, we present empirical observations on the behavior of attention matrices in a ViT model trained using softmax attention. We use the same setup as in (Dosovitskiy et al., 2021) to train a 6-layer transformer model with 6 attention heads in each of the layers on the Imagenet-1k dataset. We consider a typical input to the model and in Figures 2, 3 and 4, we present the properties of the attention weights matrix across different layers and attention heads in the model.

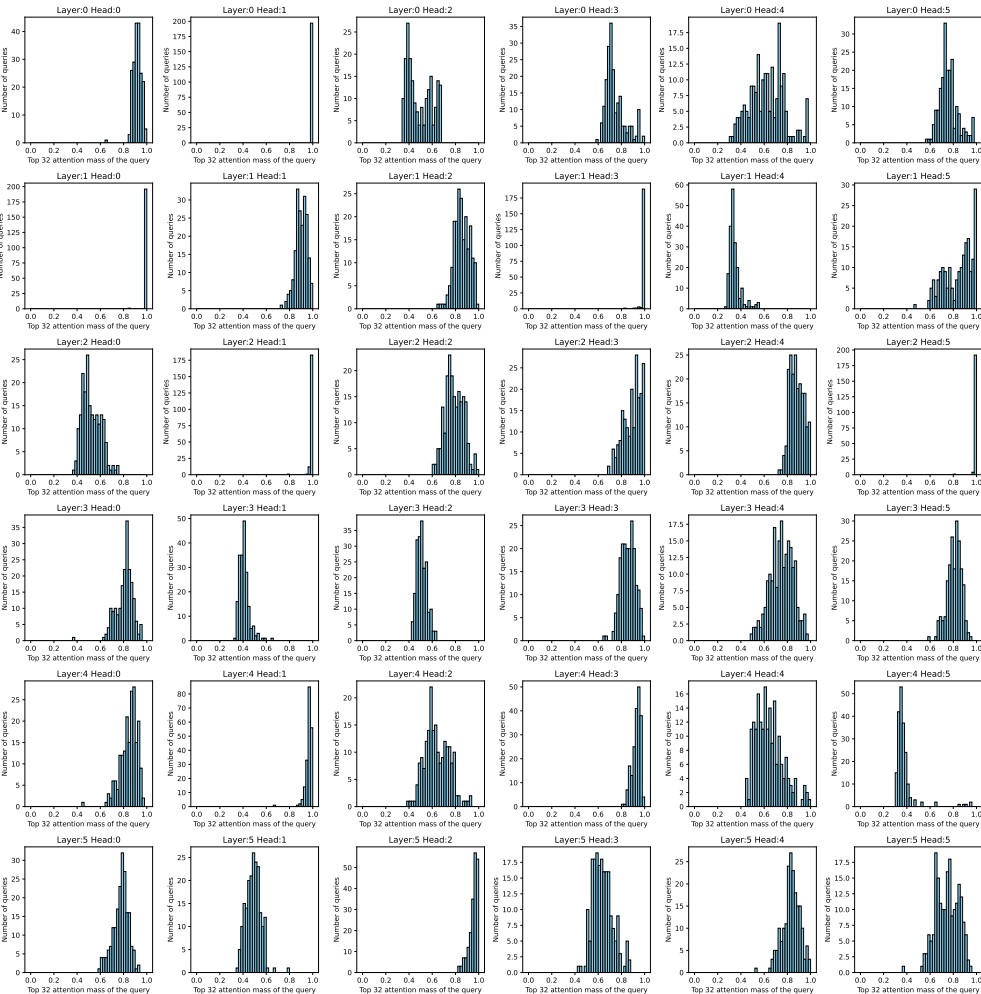

Figure 2: Histograms of top-32 attention weights. Each of the histograms plots the distribution of the sum of top-32 attention weights for query tokens. We note that at many attention heads, for many query tokens, the largest 32 attention weights constitute a significant fraction of the total attention weight.

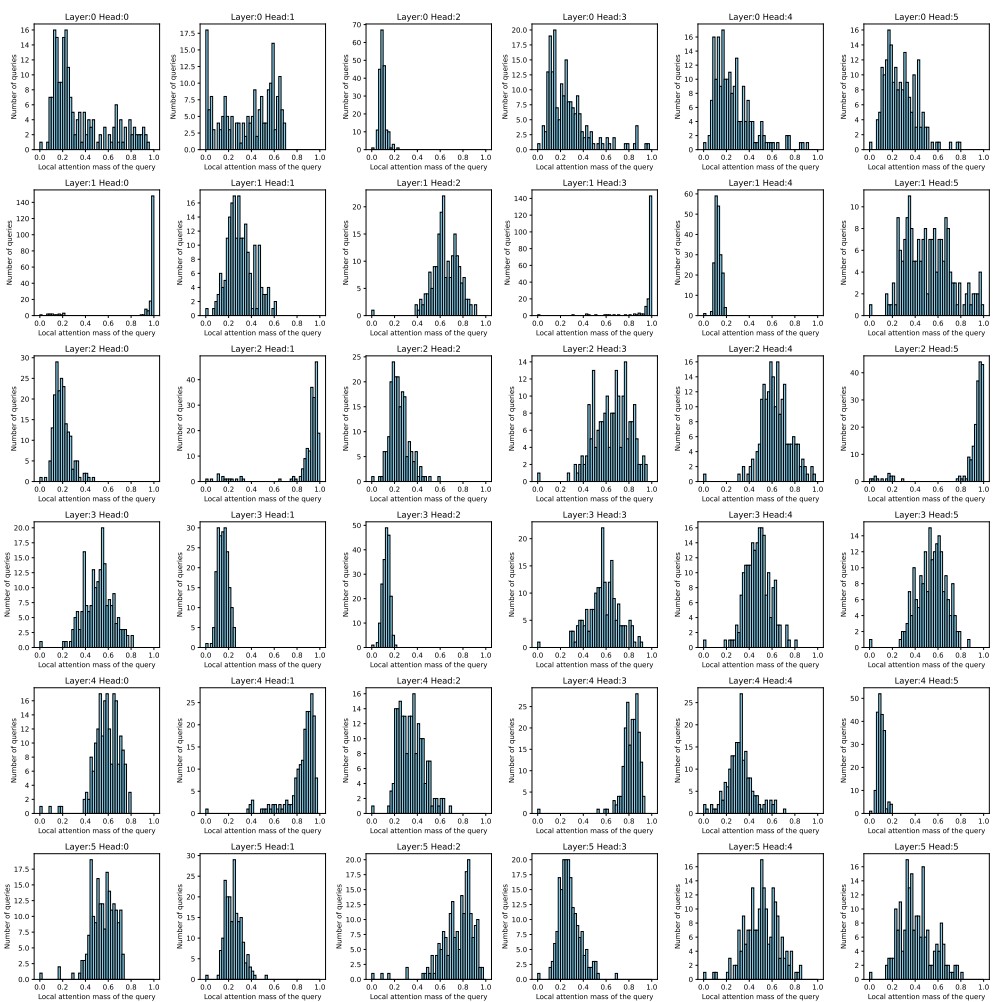

Figure 3: Histograms of attention weights captured by local tokens. Each of the histograms plots the distribution of attention weight captured by keys within a Manhattan distance of 3.

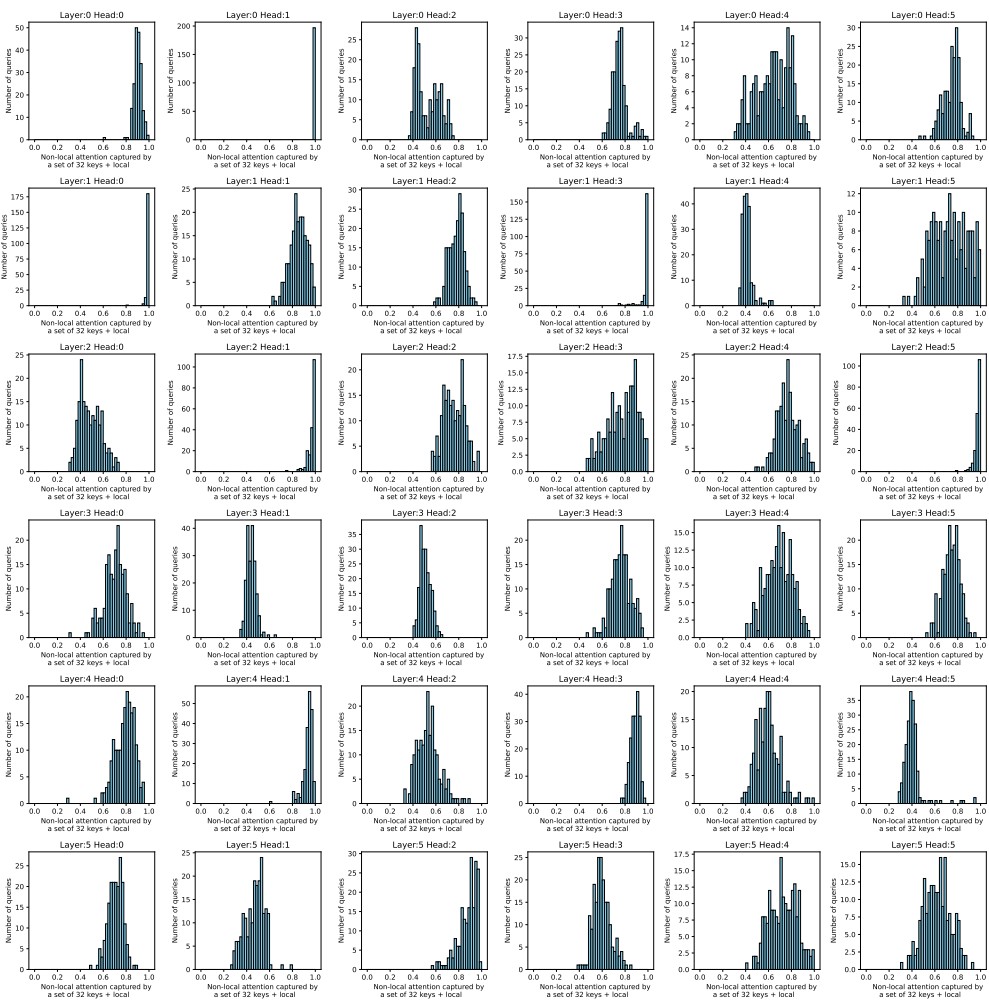

Figure 4: Histograms of Attention weights captured by the *important keys* along with *local tokens*. As discussed in Section 5.1, *important keys* are defined as the 32 keys capturing the largest attention weight at a given attention head. The histogram shows that at a large number of attention heads, the important keys together with local keys are able to capture a significant fraction of the attention weight for a large number of queries.

# 7 APPENDIX: OVERVIEW OF RECENT RELATED WORK FOR REDUCING COMPUTATIONAL COMPLEXITY

Zaheer et al. (2020) propose Big Bird, a sparse attention mechanism that combines global and local attention to handle longer sequences in transformers. While Big Bird employs a fixed sparsity pattern, our work introduces a data-dependent sparsity approach by identifying a universal set of keys based on leverage scores. These approaches could be seen as complementary: Big Bird provides a general framework for handling long sequences, while our method offers a more fine-grained approach for identifying the most relevant information in the attention matrix. Combining these techniques could be an interesting direction for future work.

Song et al. (2024) address the computational challenge of solving attention kernel regression problems, where the matrix exponential of the Gram matrix is the kernel. They propose using a pre-conditioner to accelerate the solution of these regression problems. While their work focuses on a specific formulation of the attention mechanism as a regression task, our work provides a more general method of approximating the attention matrix by identifying a universal set of important keys. Our approach could potentially complement their work by providing a reduced set of keys, which could further speed up their pre-conditioned solver.

Gao et al. (2023) propose a fast optimization approach for training single-layer attention networks in LLMs. They reformulate the attention mechanism using tensor and SVM tricks to achieve a training time comparable to matrix multiplication. While they modify the training process itself to improve efficiency, our work focuses on identifying a core set of important keys, which can be used to speed up both inference and training. Both their approach and ours could potentially be used in conjunction with learned leverage scores during training. They could incorporate learned leverage scores into their tensor-based optimization framework, while we could use learned leverage scores to construct our universal sets. Exploring the interplay between these techniques could be an interesting avenue for future research.

KDEFormer (Zandieh et al. (2023)) is a precursor to HyperAttention that uses fast kernel density estimation algorithms to approximate the softmax attention. These algorithms efficiently estimate the normalization factor for each row of $\exp(QK^T)$, as well as the sampling probabilities used in its approximate matrix multiplication. However, KDEFormer requires at least $n^{1.173}$ time and relies on assumptions such as bounded diameter datasets and small stable rank to achieve its strongest results. In contrast, our method directly identifies the most important columns with theoretical guarantees and runs in linear time with respect to $n$, without requiring these assumptions. Furthermore, as discussed in our work, the universal set we identify can be used as input to HyperAttention, effectively reducing its runtime by focusing its computation on the most relevant tokens. This could potentially lead to both faster and more accurate attention mechanisms.

