# OpenReview forum: "LevAttention: Time, Space and Streaming Efficient Algorithm for Heavy Attentions"
_ICLR.cc/2025/Conference — ICLR 2025 Poster_

### Official Review · Reviewer_tuuu · 2024-11-01

**Soundness:** 4
**Presentation:** 4
**Contribution:** 3
**Rating:** 6
**Confidence:** 3

**Summary:**

In this paper, the authors study the attention computation problem: given the query, key, and value matrices $Q, K, V \in \mathbb{R}^{n \times d}$, the goal is to output $D^{-1} A V$, where $A = \exp(QK^\top/d) \in \mathbb{R}^{n \times n}$ and $D = \mathrm{diag}(A {\bf 1}_n) \in \mathbb{R}^{n \times n}$ is a diagonal matrix. Exactly computing the product of $D^{-1}$, $A$, and $V$ requires $O(n^2 d)$ time, which can become very large when the length of the input token $n$ is large, so in this work, the authors present a novel theoretical framework and algorithm for efficiently identifying the large attention scores (the entries of the attention matrix $A$), while maintaining the linear time complexity with respect to $n$. Additionally, this paper finds the large attention scores for a large class of function $f$ in $A = f(QK^\top/d) \in \mathbb{R}^{n \times n}$, which includes all $f(x) = |x|^p$.

**Strengths:**

1. This paper gives a very solid theoretical foundation showing the existence and properties of universal sets and finds all of the large attention scores with size independent of sequence length. It applies the techniques from randomized numerical linear algebra to study the attention problem.

2. The theoretical results do not rely on any restrictive assumptions and are applicable to a wide variety of functions, including all $f(x) = |x|^p$. The algorithm can be computed in streaming and parallel settings.

3. The experimental results support the theoretical findings by considering a pre-trained ViT model. Moreover, the model quality has more than 90 percent of the accuracy of the full softmax attention when only selecting the top 32 keys.

**Weaknesses:**

1. This paper does not include an overview of the recent works, which might make it hard for readers outside this field to understand the importance of the contributions. Works like [ZGD+20] develop the sparse attention, reducing the computation complexity to $O(n)$, and other works like [SYZ23, GSWY23, ZHDK23] are more related to this work: applying the numerical linear algebra techniques, including tensor trick, preconditioner, and sketching to reduce the computational complexity.

2. The experiment only focuses on the pre-trained ViT model. This paper does not study any other language models. Therefore, it is hard to see whether or not the framework developed in this paper can be generalized to more language models.

[ZGD+20] Manzil Zaheer, Guru Guruganesh, Kumar Avinava Dubey, Joshua Ainslie, Chris Alberti, Santiago Ontanon, Philip Pham et al. "Big bird: Transformers for longer sequences." NeurIPS'20.

[SYZ23] Zhao Song, Junze Yin, and Lichen Zhang. "Solving attention kernel regression problem via pre-conditioner." AISTATS'24.

[GSWY23] Yeqi Gao, Zhao Song, Weixin Wang, and Junze Yin. "A fast optimization view: Reformulating single layer attention in llm based on tensor and svm trick, and solving it in matrix multiplication time." Preprint'23.

[ZHDK23] Amir Zandieh, Insu Han, Majid Daliri, and Amin Karbasi. "Kdeformer: Accelerating transformers via kernel density estimation." ICML'23.

**Questions:**

1. $\epsilon$ is used to determine the large attention scores. How can we decide the value of $\epsilon$?

---

> ### Author Response · Authors · 2024-11-26
> **Response to Reviewer tuuu**
>
> We thank the reviewer for their comments. We address the main ones below:
>
> -This paper does not include an overview of the recent works, which might make it hard for readers outside this field to understand the importance of the contributions. Works like [ZGD+20] develop the sparse attention, reducing the computation complexity to $O(n)$, and other works like [SYZ23, GSWY23, ZHDK23] are more related to this work: applying the numerical linear algebra techniques, including tensor trick, preconditioner, and sketching to reduce the computational complexity.
>
> We have now cited all of these works and gave a more comprehensive discussion in our updated draft - please see the second Appendix, Section 7, with a comparison of our work to each of these works. We would like to mention that due to the hardness of Alman and Song, for softmax with the exponential function and without any assumptions, it is impossible to obtain a high accuracy approximation to the attention matrix in subquadratic time. Thus, the works above come with different assumptions.
>
> -The experiment only focuses on the pre-trained ViT model. This paper does not study any other language models. Therefore, it is hard to see whether or not the framework developed in this paper can be generalized to more language models.
>
> In this paper, we focus on non-causal attention and define the universal set with respect to this. Many large language models can significantly benefit from this approach, e.g., "Lookahead When It Matters: Adaptive Non-causal Transformers for Streaming Neural Transducers" in ICML, 2023. We leave the application of these ideas to language models with causal masking as future work. There are potential challenges here in order to enforce causality. One possibiity is to use online leverage scores rather than leverage sores to ensure past tokens do not depend on future tokens through the leverage score computation.
>
>
> -$\epsilon$ is used to determine the large attention scores. How can we decide the value of $\epsilon$?
>
> One thing to note is the size of our universal set is $O(d/\epsilon)$ for $p = 2$ (and $O(d^{p/2}/\epsilon)$ in general), and thus for any $\epsilon \gg d/n$, we obtain a reduction the total number of columns of the attention. As the context length $n$ is typically much larger than the dimension $d$, this results in a significant column reduction while still finding even only moderately heavy entries. A common practice is to start with a small value for $\epsilon$ and gradually increase it while monitoring the impact on accuracy.

---

> > ### Comment · Reviewer_tuuu · 2024-11-26
> >
> > I truly appreciate the response provided by the authors and am satisfied with the rebuttal.
> >
> > One minor point:
> >
> > Regarding the first weakness, I intended to suggest that it would be beneficial to provide a comprehensive overview of recent works on the attention computation problem. This would help readers unfamiliar with the field to better understand the position of this paper within the broader context of existing research. In the revised version, the authors present a very in-depth comparison between this work and the four papers I mentioned, which is also highly valuable. If the authors are willing to further enhance the paper, I believe including a more comprehensive review of recent works would be a good direction. If not, I still think the current version is sufficiently strong.

---

> > > ### Author Response · Authors · 2024-11-27
> > >
> > > Thanks for the feedback and the further suggestion! We are happy to include a more comprehensive review of recent works to further enhance the paper.

---

### Official Review · Reviewer_n1PT · 2024-11-02

**Soundness:** 3
**Presentation:** 3
**Contribution:** 3
**Rating:** 5
**Confidence:** 3

**Summary:**

The attention mechanism is a bottleneck for transformers in terms of efficiency. Standard method of attention mechanism takes quadratic time with respect to the input, where the main issue is to calculate the n by n attention matrix A. This paper studies a class of attention mechanism where we replace f(x) = exp(x) in the vanilla attention with f(x) = |x|^p for some p, and show that regardless of the input data, there exists a set of columns of A that include all the large entries. Moreover, one can find this set efficiently. As a result, one can hope to approximate A by retaining only these large entries efficiently. The paper then generalizes the results to streaming and parallel settings.

**Strengths:**

1. Speeding up/approximating the attention mechanism is an interesting topic to study.
2. It is natural to think that as n (sequence length) grows larger, there will only be a subset of keys that are similar, i.e. have larger inner product, with the queries. This paper gives a theoretical analysis on how large such a subset can be and how we can find it.
3. This paper is clearly written and the mathematical contents are interesting.

**Weaknesses:**

1. The paper mentions that the standard attention mechanism cannot be well-approximated unless SETH is false (Alman & Song 2023), so I guess the point is that we are hoping for a good/efficient approximation for other attention variants. Although f(x) = |x|^p has been proposed and used in previous work (PolySketchFormer, TensorSketch), I don’t think they are the “gold standard” for attention, and therefore the motivation of studying this polynomial attention is not so clear to me.
2. Even if we can find all the columns containing all the large entries, does that imply a reasonable approximation after we multiply the attention matrix with the value matrix V? In addition, if we consider multiple self-attention layers, even though we can find the important columns of all attention matrices, it is not clear to me whether we can still have any kind of guarantee on the final output.

Overall I think the problem that this paper proposes is an interesting problem in numerical linear algebra, but I am not really convinced about its impact on transformer theory.

**Questions:**

The result in this paper, as the authors stated, applies to any Q. It seems like this is because of the way f-sensitivity is defined (by taking the sup over all y). In practice Q might be structured (I think there is a shared-QK transformer where Q and K are identical), and I wonder if we can say anything about it.

---

> ### Author Response · Authors · 2024-11-26
> **Response to Reviewer n1PT**
>
> We thank the reviewer for their comments. We address the main ones below:
>
>
> - The result in this paper, as the authors stated, applies to any $Q$. It seems like this is because of the way $f$-sensitivity is defined (by taking the sup over all $y$). In practice $Q$ might be structured (I think there is a shared-$QK$ transformer where $Q$ and $K$ are identical), and I wonder if we can say anything about it.
>
> Our structural results apply regardless of whether $Q = K$ or not, but we think the reviewer is asking if one can perhaps obtain a small universal set for softmax itself when $Q = K$. Unfortunately with softmax, the unbounded nature of the exponential function means that in certain cases, there cannot exist a universal set of size less than $n$, where $n$ is the total number of columns in the matrix. To see this, suppose $K = Q$ is a random $n \times d$ sign matrix, where $d = C \ln n$, for a large constant $C > 0$. Then the diagonal entries of $KQ^T$ equal $C \ln n$, whereas by standard concentration bounds, with high probability the off-diagonal entries are simultaneously all at most $\sqrt{C \ln n} \sqrt{\log n}$ in absolute value. Thus, softmax$(\exp(KQ^T))$ will have diagonal entries that are close to $1$, and off-diagonal entries that are at most $1/n$. Consequently, the only universal set of softmax$(\exp(KQ^T))$ is the entire set of n columns.
>
> - The paper mentions that the standard attention mechanism cannot be well-approximated unless SETH is false (Alman & Song 2023), so I guess the point is that we are hoping for a good/efficient approximation for other attention variants. Although $f(x) = |x|^p$ has been proposed and used in previous work (PolySketchFormer, TensorSketch), I don’t think they are the “gold standard” for attention, and therefore the motivation of studying this polynomial attention is not so clear to me.
>
> As you note, Softmax with the exponential function unfortunately requires quadratic time. Even now, for very long contexts, it is too expensive in terms of both running time and cost to spend $n^2$ time on state of the art models in every head and in every layer, and this will only become more problematic as $n$ grows larger. Our work thus fits into a growing body of methods for bypassing the quadratic time barrier.
>
> In terms of specific motivation for looking at $f(x) = |x|^p$, we first give both empirical and theoretical motivation. Empirically, our experiments show that leverage-score based pruning ($f(x) = x^2$ based) is effective. From the theory side, in addition to our main results, we formulate a stochastic model of $Q$ and $K$ and show that under this model, $f(x) = |x|^p$ yields strong results. We make a beginning with our planted model which we think will inspire the study of more detailed models.
>
> - Even if we can find all the columns containing all the large entries, does that imply a reasonable approximation after we multiply the attention matrix with the value matrix $V$? In addition, if we consider multiple self-attention layers, even though we can find the important columns of all attention matrices, it is not clear to me whether we can still have any kind of guarantee on the final output.
>
> We are currently not aware of any work that guarantees any end-to-end guarantees for a full transformer, i.e., with multiple self-attention layers. Recent work, such as HyperAttention, provides end-to-end guarantees when multiplying by $V$ in a single layer, but it requires multiple assumptions and it is not clear if these assumptions always hold in practice. That said, HyperAttention works by separately finding the heavy attention scores and estimating the contribution from the light attention scores, and so we can use our universal set as a preprocessing step to HyperAttention, since any heavy attention must be inside our universal set. We can also adjust eps in practice, i.e., even $\epsilon = 1/\sqrt{n}$ gives a universal set of size $n^{1/2} d$, which can result in a significant savings in runtime for HyperAttention.
>
> - Overall I think the problem that this paper proposes is an interesting problem in numerical linear algebra, but I am not really convinced about its impact on transformer theory.
>
> We believe the concepts of universal keys and the planted model could lead to new theoretical bounds on the complexity of attention mechanisms or inspire new approximation algorithms with provable guarantees.

---

> > ### Author Response · Authors · 2024-12-02
> > **Follow up to Reviewer n1PT**
> >
> > Dear Reviewer n1PT,
> >
> > Thank you again for your review.
> >
> > We believe we have addressed your question about structured Q, and concerns regarding polynomial attention and end-to-end approximation. If any of your questions or concerns have not been addressed, could you please let us know before the end of the discussion phase?
> >
> > Many thanks,
> > The authors

---

### Official Review · Reviewer_k6Cn · 2024-11-03

**Soundness:** 3
**Presentation:** 3
**Contribution:** 2
**Rating:** 6
**Confidence:** 3

**Summary:**

This paper develops efficient algorithms to identify significant attention scores in transformer architectures without computing the full attention matrix. The key theoretical contribution is proving the existence of a small "universal set" of keys, independent of sequence length, that captures all large attention scores for any query. The authors provide efficient algorithms to find this set and compute attention scores in streaming and distributed settings. The work provides both theoretical guarantees and practical benefits for improving transformer efficiency.

**Strengths:**

1. Rigorous mathematical proofs for existence and size of universal sets
2. Straightforward integration with existing transformer architectures
3. Strong empirical results on vision transformers

**Weaknesses:**

1. No evaluation on (large) language models. Limited ablation studies on prediction quality

2. Limited ablation studies on prediction quality

**Questions:**

1. What are the trade-offs between exact and approximate computation of attention scores?

2. Can the approach be parallelized effectively across multiple GPUs (Tensor Parallel)?

3. What is the impact on training time and convergence?

---

> ### Author Response · Authors · 2024-11-26
> **Response to Reviewer k6Cn**
>
> We thank the reviewer for their comments. We address the main ones below:
>
> - Evaluation on Large Language Models
>
> In this paper, we focus on non-causal attention and define the universal set with respect to this. Many large language models can significantly benefit from this approach, e.g., "Lookahead When It Matters: Adaptive Non-causal Transformers for Streaming Neural Transducers" in ICML, 2023. We leave the application of our ideas to improve the efficiency of language models with causal masking as future work. One possibility is to use online leverage scores rather than leverage sores to enforce causality.
>
> -Limited ablation studies on prediction quality
>
> We have performed experiments with local tokens added along with the universal set of tokens but have observed that the performance doesn't improve much over the end-to-end accuracies that we report in Table 1. This shows that while adding local tokens improves the amount of attention weight captured, it does not seem to improve the accuracy in the image classification task. We are happy to add more ablation study details, such as varying the number of local tokens or different local token selection strategies.
>
> -What are the trade-offs between exact and approximate computation of attention scores?
>
> The short answer, ignoring polynomial factors in epsilon and logarithmic factors in $n$, the time for exact computation is $nd^2$ while for approximate computation it is $nd$.
>
> In more detail, given a key matrix $K$, one can compute its SVD $K = U \Sigma V^T$, where $U$ is $n \times d$, $\Sigma$ is $d \times d$, and $V^T$ is $d \times d$. This takes $O(nd^2)$ time and is done once for a given key matrix $K$. Given a query $q$, we can compute the normalization factor for the $q$-th row of the attention matrix as $\|Kq\|_2^2$. Importantly, this equals $\|\Sigma V^T q\|_2^2$ since $U$ has orthonormal columns. Consequently, for each query $q$ (row of attention matrix), we can compute its normalization factor *exactly* in only $O(d^2)$ time. As there are only $O(d/\epsilon)$ columns of the attention matrix that could contain an epsilon-heavy entry, we just need to evaluate $\langle q, k \rangle^2$ for each key $k$ corresponding to one of these $O(d/\epsilon)$ columns.  This takes $O(d^2/\epsilon)$ time. Thus, in $O(d^2/\epsilon)$ time we can compute all heavy entries in a single row of the attention matrix exactly, and in $O(nd^2/\epsilon)$ time all heavy attention scores in the entire attention matrix exactly.
>
> This can be sped up using sketching. By using the Johnson-Lindenstrauss transform $R$ with $O(\log n /\epsilon^2)$ rows, we have $\|RKq\|_2^2 = (1+-\epsilon) \|Kq\|_2^2$ simultaneously for all $n$ queries. Further, $R \cdot K$ can be computed in $\tilde{O}(nd)$ time for any $\epsilon > 1/\sqrt{d}$ using fast Johnson Lindenstrauss transforms. As $R \cdot K$ is a small matrix, for each of $n$ new queries $q$ we compute $\|RKq\|_2^2$ in $O((\log n) d/\epsilon^2)$ time, so $O(nd (\log n)/\epsilon^2)$ time in total to compute the normalization factor for all queries up to a multiplicative $1+\epsilon$ factor.
>
> For the entries of $\exp(QK^T)$ before normalization, we can find a superset $S$ of $O(d/\epsilon)$ columns containing the large leverage scores in $\tilde{O}(nd + d^{\omega})$ time using sketching, where $\omega  < 2.37$ is the exponent of fast matrix multiplication (one could set $\omega = 3$ since the context length $n$ may be much larger than $d$). See "Fast Algorithm for Finding $U$" in Section 1.1 for references. Now we only need to compute the large entries of $Q \cdot S$, where $Q$ is the $n \times d$ query matrix, and $S$ is a $d \times O(d/\epsilon)$ matrix with the keys corresponding to the universal set. We can again use sketching to instead compute $Q \cdot S \cdot T$, where $T$ is an $O(d/\epsilon) \times O((\log n)/\epsilon)$ CountSketch matrix which can be used to find the heavy entries in each row of $Q \cdot S$. Importantly we compute $S \cdot T$ first in $d^2 \textrm{poly}((\log n)/\epsilon)$ time, at which point $S \cdot T$ is a $d \times O((\log n)/\epsilon)$ sized matrix, and we can compute $Q \cdot (ST)$ in only $O(nd (\log n)/\epsilon)$ time. We can then divide each row by the normalizations found in the previous paragraph.
>
> To summarize, while approximate computation reduces time complexity from $nd^2$ to $nd \cdot \textrm{poly}((\log n)/\epsilon)$, it introduces a small approximation error controlled by epsilon.

---

> > ### Author Response · Authors · 2024-11-26
> > **Continued Response to Reviewer k6Cn**
> >
> > - Can the approach be parallelized effectively across multiple GPUs (Tensor Parallel)?
> >
> > For non-causal attention, the usual way to speed it up is to shard over the token dimension and then do an all-gather over the key shards so that each of the machines ends up with all of the keys and a portion of the queries, and each of the machines then computes attention on their local query chunk. One way to speed up attention in such a setting using our technique is to first do a subset selection for each of the key chunks and only do an all-gather over the selected key chunks. We can thus further reduce the number of keys on all of the machines by an additional round of leverage score selection on the subset of keys. This decreases the communication requirements for the all-gather operation and the computational requirements for attention on query chunks for each of the machines.
> >
> > - What is the impact on training time and convergence?
> >
> > In some of our experiments, we use existing pretrained models and use the leverage score attention only at inference time and hence the technique needs no additional training time. In other experiments, we train the models from scratch using the leverage score attention mechanism, which does have a larger step time compared to an optimized softmax attention implementation. For this paper, we did not make significant effort to optimize the training time of `leverage score based attention’ since our main focus is on studying the quality and accuracy aspects of the proposed mechanism.

---

> > > ### Comment · Reviewer_k6Cn · 2024-11-26
> > >
> > > Thank you for replying my questions. After reading the comments and replies from all other reviewers and the authors, I think this is a good work on improving the efficiency for heavy attention algorithm. I will keep my current score of 6. I hope this algorithm can be implemented under a larger scenario and used by more people in the future.

---

### Official Review · Reviewer_MVJp · 2024-11-03

**Soundness:** 3
**Presentation:** 2
**Contribution:** 2
**Rating:** 6
**Confidence:** 3

**Summary:**

This paper studies efficient algorithms for attention computation in transformer models. In particular for query $Q$ and key $K$ matrices in $\mathbb{R}^{n\times d}$, and function $f$ applied to each entry of $QK^T$ (for eg. $f=x^p$ for some $p>0$), this paper studies efficient algorithms to approximately compute $f(QK^T)$ followed by a row-normalization. The main contribution of the paper is to show that for any $\epsilon>0$ there exist a subset $U\subset[n]$ of keys, that is the rows of the key matrix $K$, such that for any query $q$, that is a row of query matrix $Q$, if the attention score of $q$ with $k_i$ after normalization is greater than $\epsilon$ then $i\in U$. The size of this set $U$ is $(\sum_{i\in [n]}\sigma_i^f(K))/\epsilon$ where $\sigma_i^f$ is the $i^{th}$ $f$-sensitivity score of $K$. For eg. when $f=x^2$ these sensitivities are nothing but the $\ell_2$ leverage scores of $K$. The set $U$ is independent of $Q$ and thus can be used to compute attention scores with any $Q$ in the future for a fixed $K$. Moreover using fast algorithms developed in the literature for computing $f$-sensitivities for a broad class of functions $f$, the set $U$ can be computed efficiently. For eg for any constant $p$, for $x^p$ the set $U$ can be computed in time $nnz(K)+poly(d/\epsilon)$ using the input sparsity time algorithm of Cohen and Peng for computing $\ell_p$ leverage scores.

**Strengths:**

The main strength of the paper is to apply tools from randomized numerical linear algebra to naturally arrive to the conclusion that if the attention score of a query is more than $\epsilon$ with any particular key, then simply by definition it implies that the $f$-sensitivity (or leverage score for the special case when $f=x^2$) will also be more than $\epsilon$. This directly implies that capturing all the keys of $K$ with sensitvities higher than $\epsilon$ suffices to prove their result. Thus I think the main strength of this paper is develop this connection in more detail, and prove results in various computational settings such as streaming and distributed settings regarding how to efficiently compute this set of universal keys, which I think is a good contribution to the literature.

**Weaknesses:**

One aspect I would want to get more clarity on is in the experimental section. The authors do arrive at the conclusion that the mass of practical attention matrices for each query can be captured by a small set of universal keys plus a set of local tokens for that specific query. However if this is the case, then algorithms for computing the set of local tokens should also be considered on top of computing the set of universal keys. For eg. it may be the case that in natural language applications, there are sentences in which for each token, it has a high correlation with a few local tokens in a small window around it. In this case, the leverage scores/sensitivities of each key may be large resulting in a large universal set, and the attention matrix has a high rank. However, the attention matrix can still be efficiently approximated with a sparse matrix as it essentially amounts to computing, for each token, the small set of local tokens with whom it has a high correlation.

**Questions:**

The main question I have is around the weakness that I brought up regarding the experimental section - that is how would the case be handled when $K$ has high rank and every key has high sensitivity ? Morever in the experimental section have the authors considered evaluation with other methods which are popularly used in attention approximation such as flash attention ?

---

> ### Author Response · Authors · 2024-11-26
> **Response to Reviewer MVJp**
>
> We thank the reviewer for their comments. We address the main ones below:
>
> - The main question I have is around the weakness that I brought up regarding the experimental section - that is, how would the case be handled when $K$ has high rank and every key has high sensitivity ?
>
> Our theoretical results give a small universal set size (of size roughly $d^{p/2}/\epsilon$) when applied to activation functions that have at most polynomial growth, and hold for any possibly worst case attention matrix.
>
> Unfortunately with softmax, the unbounded degree of the exponential function means that in certain cases, there provably cannot exist a universal set of size less than $n$, where $n$ is the total number of columns of the attention matrix. Here is a simple example which illustrates that the universal set can have cardinality $n$: suppose $Q = K$ is a random $n \times d$ sign matrix, where $d = C \ln n$, for a large constant $C > 0$. Then the diagonal entries of $QK^T$ equal $C \ln n$, whereas by standard concentration bounds, with high probability the off-diagonal entries are simultaneously all at most $\sqrt{C \ln n} \sqrt{\log n}$ in absolute value. Thus, softmax$(\exp(QK^T))$ will have diagonal entries that are close to $1$, while all off-diagonal entries will be $O(1/n)$. Consequently, the only universal set of softmax$(\exp(QK^T))$ is the entire set of n columns.
>
> The example in the previous paragraph is worst-case, and the goal of our experiments was to investigate whether the insights gained from polynomial attention, where we have stronger theoretical guarantees, carry over to the more widely used softmax attention for real-world, practical use cases. Despite the worst case theoretical example for softmax given above, we observed a promising trend in the experiments: even with a relatively small universal set comprised of keys with high leverage scores (corresponding to polynomial activation with $p = 2$ in our theoretical results), we were able to capture a significant portion of the {\bf softmax attention} mass for each token by taking the universal set that we found for $p = 2$.
>
> We also performed experiments with local tokens added along with the universal set of tokens but have observed that the performance does not improve much over the end-to-end accuracies that we report in Table 1. This shows that while adding local tokens improves the amount of attention weight captured, it does not seem to improve the accuracy in the image classification task.
>
> Ultimately, even if the attention matrix has high rank due to these local dependencies, the ability to approximate it with a sparse representation using both universal and local tokens offers significant computational advantages compared to standard softmax attention.
>
> - Moreover in the experimental section have the authors considered evaluation with other methods which are popularly used in attention approximation such as flash attention ?
>
> Our work is somewhat orthogonal to FlashAttention, which is an efficient hardware-based implementation of the quadratic-time softmax attention. Our goal is to understand structurally how much of the heavy attention mass can be captured using a small universal set of columns, together with a few local tokens. The universal set that we find, together with local tokens, could in principle be used as a preprocessing step for other attention mechanisms.
>
> We do not claim that our implementations of leverage score based attention are highly optimized, and therefore do not expect to be faster than FlashAttention at the context lengths we consider. One should note though that as context lengths become larger, even a highly optimized quadratic-time implementation such as FlashAttention will be prohibitive, which motivates the design of subquadratic time algorithms.

---

> > ### Comment · Reviewer_MVJp · 2024-12-01
> >
> > Thanks a lot for the main clarification, basically using a polynomial activation function helps to circumvent the issue faced by the softmax attention.  I am satisfied with the authors responses, and thank them for their work and the responses.

---

### Meta-Review · Area_Chair_Bv8k · 2024-12-15

**Metareview:**

The paper considers a new algorithm for attention computation under polynomial activation function. There is a need to understand the efficiency of attention mechanisms given their importance in transformer, and the theoretical results here are strong enough that I recommend acceptance.

**Additional Comments On Reviewer Discussion:**

Rebuttal was useful and helped to come up with the final recommendation.

---

### Decision · Program_Chairs · 2025-01-22

Accept (Poster)